# Thriving in Wetlands: Ecophysiology of the Spiral-Shaped Methanotroph *Methylospira mobilis* as Revealed by the Complete Genome Sequence

**DOI:** 10.3390/microorganisms7120683

**Published:** 2019-12-11

**Authors:** Igor Y. Oshkin, Kirill K. Miroshnikov, Olga V. Danilova, Anna Hakobyan, Werner Liesack, Svetlana N. Dedysh

**Affiliations:** 1Winogradsky Institute of Microbiology, Research Center of Biotechnology of the Russian Academy of Sciences, Moscow 119071, Russia; ig.owkin@gmail.com (I.Y.O.); infon18@gmail.com (K.K.M.); vinnigo@gmail.com (O.V.D.); 2Max-Planck-Institut für Terrestrische Mikrobiologie, D-35043 Marburg, Germany; anna.hakobyan@mpi-marburg.mpg.de (A.H.); liesack@mpi-marburg.mpg.de (W.L.)

**Keywords:** spiral-shaped methanotrophs, *Methylospira mobilis*, wetlands, complete genome sequence, comparative genome analysis, signal transduction systems, motility genes, nitrogenase, environmental adaptations

## Abstract

*Candidatus* Methylospira mobilis is a recently described spiral-shaped, micro-aerobic methanotroph, which inhabits northern freshwater wetlands and sediments. Due to difficulties of cultivation, it could not be obtained in a pure culture for a long time. Here, we report on the successful isolation of strain Shm1, the first axenic culture of this unique methanotroph. The complete genome sequence obtained for strain Shm1 was 4.7 Mb in size and contained over 4800 potential protein-coding genes. The array of genes encoding C_1_ metabolic capabilities in strain Shm1 was highly similar to that in the closely related non-motile, moderately thermophilic methanotroph *Methylococcus capsulatus* Bath. The genomes of both methanotrophs encoded both low- and high-affinity oxidases, which allow their survival in a wide range of oxygen concentrations. The repertoire of signal transduction systems encoded in the genome of strain Shm1, however, by far exceeded that in *Methylococcus capsulatus* Bath but was comparable to those in other motile gammaproteobacterial methanotrophs. The complete set of motility genes, the presence of both the molybdenum–iron and vanadium-iron nitrogenases, as well as a large number of insertion sequences were also among the features, which define environmental adaptation of *Methylospira mobilis* to water-saturated, micro-oxic, heterogeneous habitats depleted in available nitrogen.

## 1. Introduction

Aerobic methane oxidizing bacteria or methanotrophs are a group of bacteria characterized by the unique ability to utilize methane (CH_4_) as a sole source of energy [1,2,3,4]. This ability is due to the possession of methane monooxygenase (MMO) enzymes, which exist in particulate (pMMO) and soluble (sMMO) forms and catalyze the critical step of aerobic methanotrophic metabolism, i.e., oxidation of methane to methanol [5,6]. The currently described diversity of aerobic methanotrophs includes representatives of the *Gamma*- and *Alphaproteobacteria* (also known as type I and type II methanotrophs, respectively) and the *Verrucomicrobia* [7]. Although methanotrophic representatives of the candidate phylum NC10 also possess MMO, these bacteria occur in anoxic habitats and employ an intra-aerobic pathway of CH_4_ oxidation [8]. Methanotrophic *Verrucomicrobia* are restricted to acidic geothermal habitats [9], while proteobacterial methanotrophs occur in various terrestrial, freshwater, and marine ecosystems, typically at oxic–anoxic interfaces, where both methane and oxygen are available [1,10].

Methanotrophic bacteria display an impressive range of phenotypes and habitat-specific adaptations. One of the most surprising, recently described phenotypes of proteobacterial methanotrophs is represented by *Candidatus* Methylospira mobilis, a spiral-shaped, micro-aerophilic methanotroph, which is capable of rapid motility in water-saturated, heterogeneous habitats with high microbial biofilm content [11]. As suggested by culture-independent studies, *Methylospira*-like methanotrophs are widely distributed in various northern environments, such as wetlands, organic soils, and lake sediments. These bacteria were successfully cultivated in the laboratory with methane as the only growth substrate. They displayed CH_4_-tactic responses, thereby being capable of fast moving through semisolid agar medium under methane and oxygen concentration gradients. Despite all purification efforts, this methanotroph could not be obtained in pure culture and, therefore, was assigned the status *Candidatus* [11]. We, however, continued our purification work, which finally resulted in obtaining the first axenic culture of this unique methanotroph, strain Shm1. This gave us the possibility to determine its complete genome sequence and to examine the range of genome-encoded traits and environmental adaptations.

Based on 16S rRNA gene phylogeny, the closest described relatives of strain Shm1 are *Methylococcus capsulatus* Bath (94.06% sequence similarity) and *Methyloterricola oryzae* 73a^T^ (95.17%)*. Mc. capsulatus* Bath is probably the best characterized aerobic methanotroph, which has been the “workhorse” organism for researchers studying the biology of methane oxidation for over 40 years [12]. Research on its physiology and biochemistry, but also genomics, transcriptomics, proteomics, and metabolic modelling, provided extensive information on the metabolic features of this bacterium [13,14,15,16,17,18,19,20]. The first published methanotroph genome was that of *Mc. capsulatus* Bath [15]. *Methyloterricola oryzae* 73a^T^ is a mesophilic methanotroph, which was isolated from rice stems of a paddy rice field in the Philippines [21]. In contrast to *Mc. capsulatus* Bath, information on *Methyloterricola oryzae* 73a^T^ is limited and only a draft genome sequence is available for this methanotroph [22]. Therefore, we used *Mc. capsulatus* Bath as a reference for the genome analysis of strain Shm1. Notably, despite their close phylogenetic relatedness, these two methanotrophs differ dramatically in their cell morphology and phenotype. *Mc. capsulatus* Bath is a non-motile, moderately thermophilic and obligately aerobic bacterium of coccoid morphology. The highly motile, mesophilic, and micro-aerophilic strain Shm1 with its helical cells thus represents ‘an antipode’ to *Mc. capsulatus* Bath. Our comparative genome analysis was focused on genome-encoded traits that define different life strategies of these two closely related gammaproteobacterial methanotrophs. In addition, we have compared the arrays of genes related to motility, signal transduction, or adaptation to micro-oxic conditions in strain Shm1 to those in several motile gammaproteobacterial methanotrophs. 

## 2. Materials and Methods

### 2.1. Isolation Procedure

The culture of *Candidatus* Methylospira mobilis described by Danilova et al. [11] was grown in the modified liquid DSMZ medium 1181 (DNMS, [23]) containing (in gram per litre) MgSO_4_, 0.2; KNO_3_, 0.2; CaCl_2_, 0.04; KH_2_PO_4_, 0.054; Na_2_HPO_4_ × 12H_2_O, 0.143 with the addition of 0.1% (*v*/*v*) of a trace elements stock solution. Final pH of the medium was 5.8. A set of 500 mL bottles were filled to 30% capacity with this medium, sealed with rubber septa, and CH_4_ (30%, *v*/*v*) was added to the headspace using syringes equipped with disposable filters (0.22 µm). The culture obtained after two weeks of incubation under static conditions contained three cell morphotypes, including the target organism (large, spiral-shaped cells) and two satellite bacteria (small short rods and very thin helical cells). This culture was subjected to gradient centrifugation, which is an efficient approach for separating cells of different sizes [24,25]. The culture was centrifuged in a stepwise (from 5% to 50% in 5% increments) gradient of sucrose in water at 2400× *g* for 30 min. Visible band of target bacteria appeared in the range of 15–20% sucrose concentration. Aliquots of cell suspensions from this layer were used for preparing two sets of ten-fold serial dilutions in a liquid DNMS medium. One of these sets corresponded to fully aerobic conditions (120 mL flasks were filled to 20% capacity and incubated on a shaker at 120 rpm), while the other one reproduced micro-oxic conditions (500 mL bottles were filled to 60% capacity and incubated under static conditions). Growth of the target spiral-shaped methanotroph occurred only in a series of bottles incubated under static conditions and only one of the bottles contained a pure culture of these bacteria. This isolate was designated strain Shm1.

### 2.2. Cultivation in a Range of Oxygen Concentrations and Analysis of Chemotactic Properties

Since *Candidatus* Methylospira mobilis was originally described as an obligately micro-aerophilic methanotroph [11], we examined the oxygen preferences of the newly isolated strain Shm1 using two different approaches. The first approach involved cultivation of this bacterium in opposing gradients of methane and oxygen as described by Bussmann et al. [26] Briefly, glass tubes (25 mm × 12 cm) were supplied with cell suspension of strain Shm1, mixed with semisolid DNMS medium containing 0.3% (*w*/*v*) agarose and sealed with rubber septa at both ends. Lower end of the tube was supplied with 4% CO_2_, 30% CH_4_, and balance N_2_. The upper end was exposed to air. The incubation was done for two weeks at room temperature.

Another approach measured the specific growth rates of strain Shm1 in incubations with different oxygen concentrations. To create reduced oxygen conditions, a series of 120-mL flasks containing 20 mL DNMS medium were flushed with nitrogen for 2 min, after which N_2_ pressure was allowed to equalize. Sixty, forty, twenty, and 8 milliliters of nitrogen were replaced with atmospheric air to achieve a concentration of 15%, 10%, 5%, and 2% (*v*/*v*) O_2_ in the headspace. Methane was added a concentration of 10% (*v*/*v*). The concentration of oxygen was verified by using oxygen sensors (PreSens, Regensburg, Germany) attached to the glass surface inside the flasks. Growth was assessed by measuring OD_600_ after 20 h of incubation on a shaker at 100 rpm.

Experimental verification of CH_4_- and CH_3_OH-tactic responses of strain Shm1 was performed using the previously published assay [11].

### 2.3. Genome Sequencing and Assembly

Genome sequencing of strain Shm1 was performed at the Max Planck Genome Centre Cologne (MP-GCC). Genomic DNA was extracted from the biomass of strain Shm1 using the kit “Genomic DNA from food” (Macherey-Nagel, Düren, Germany). The genome was sequenced using the PacBio Sequel platform (Pacific Biosciences, Menlo Park, CA, USA). De novo assembly was done using the hierarchical genome-assembly process (HGAP3) via the SMRT Portal offered by Pacific Biosciences. The genomic sequence of *Ms. mobilis* Shm1 has been deposited in the DDBJ/NCBI/ EMBL databases (accession no. CP044205).

### 2.4. Annotation 

Protein-coding sequences (CDSs) were predicted using RAST v. 2.0 (Rapid Annotation using Subsystem Technology) [27], Prokka [28], and GhostKOALA [29]. The transfer RNA (tRNA) and rRNA genes were identified using tRNAScan-SE [30] and RNAmmer [31], respectively. Functional annotation of CDSs was performed on the basis of the results of BLASTP searches [32] against the NCBI (National Center for Biotechnology Information) nonredundant database and the KEGG (Kyoto Encyclopedia of Genes and Genomes) database [33]. Insertion sequence (IS) region search was done by ISsaga server [34]. The search for prophage elements in the genome was performed using the Phaster server [35].

### 2.5. Comparative Genomics and Phylogenetic Analysis

The genome sequence of *Mc. capsulatus* Bath was obtained from the GenBank (accession No AE017282.2). The overall similarities between the genomes of strain Shm1 and *Mc. capsulatus* Bath were estimated using average nucleotide identity (ANI) calculator and formula 2 of the Genome-to-Genome-Distance-Calculator [36]. A genome-based tree of strain Shm1 was reconstructed using the Genome Taxonomy Database and GTDB-Tk (https://github.com/Ecogenomics/GtdbTk) [37].

Functional annotation of CDSs was performed as described above. The whole-genome alignment was created with LASTZ [38]. Based on this alignment, Circos was used to generate a circular map of the two genomes and presence/absence information regarding genes of interest [39]. The GC skew was calculated with the help of GenSkew (http://genskew.csb.univie.ac.at/) using default parameters.

Genome sequences of several motile gammaproteobacterial methanotrophs, i.e., *Methylomonas methanica* MC09 (NC_015572.1), *Methylomonas* sp. LW13 (NZ_CP033381.1), *Methylobacter tundripaludum* SV96 (AEGW00000000.2) and *Methylomicrobium buryatense* 5GB1C (NZ_CP035467.1) were obtained from the GenBank for comparison of repertoires of motility- and dinitrogen fixation-related genes as well as terminal oxidases and signal transduction systems encoded in the genomes. The list of terminal oxidases and *vnfD* genes for phylogenetic analyses was compiled from the results of PSI-BLAST searches using terminal oxidases encoded in the genomes of strain Shm1 and *Mc. capsulatus* Bath as the sequence queries [32]. Sensory domains were identified by searching against Pfam [40] and Smart domain databases [41]. The approximately maximum-likelihood trees of cytochrome oxidases and VnfD were constructed using the FastTree package [42].

## 3. Results

### 3.1. Isolation of Methylospira Mobilis Shm1

Given that conventional isolation procedures such as serial dilutions and surface plating were not successful in the case of *Candidatus* Methylospira mobilis [11], a gradient centrifugation technique was applied to separate the target spiral-shaped methanotroph from cells of satellite bacteria. Due to significant differences in cell size, this separation worked well. The proportion of satellite bacteria in the obtained fraction of target cells was reduced from 5% to below 0.1%. This highly enriched cell fraction was taken for further purification work by means of multiple serial dilutions with incubations under both fully aerobic and micro-oxic conditions. This one-year-long purification effort succeeded in obtaining a desired pure culture of a spiral-shaped methanotroph, designated strain Shm1, in one of the bottles incubated under micro-oxic conditions (Figure 1A).

Analysis of the 16S rRNA gene sequence of strain Shm1 revealed that it was nearly identical (99.87% similarity, 2 mismatches over the stretch of 1498 nucleotides) to that determined for *Candidatus* Methylospira mobilis (accession number KU216206).

### 3.2. Phenotypic Characterization

Strain Shm1 was represented by spiral-shaped cells that divided by binary fission and were highly motile in the exponential growth phase. They did not form colonies on solid media and grew optimally in liquid or semisolid media under static conditions. In the experimental set-up for growth in the opposing gradients of methane and oxygen (see Section 2.2), cells of strain Shm1 developed visible bands at a depth of 35–40 mm below the air-exposed surface (Figure 1B).

In order to estimate the influence of oxygen concentrations on its growth dynamics, strain Shm1 was cultivated in the flasks containing 2%, 5%, 10%, 15%, and 19% O_2_ in the headspace (Figure 1C). Best growth was observed in the range of 2–10% O_2_, with highest specific growth rate (0.049 ± 0.006 h^−1^) detected in the incubation with 5% O_2_ in the headspace. Increasing oxygen concentration up to 15% and 19% resulted in a decrease of the specific growth rates down to 0.029 ± 0.002 and 0.025 ± 0.002 h^−1^, respectively.

The chemo-tactic responses of strain Shm1 were verified using a syringe migration assay (Figure 2A) [11]. The cells were actively migrating into the syringes containing either methane or methanol vapors in the headspace (Figure 2B). No cell migration was observed in case of control syringes containing ambient air in the headspace.

### 3.3. General Genome Features of Strain Shm1 and Genome-Based Phylogeny

A total of 713,543 PacBio reads were obtained with a mean length of 13,195 bp (4803-fold coverage of the genome). The genomic assembly obtained for strain Shm1 represented a single contig of 4.7 Mbp, with an average G+C content of 54 mol %. The chromosome contains three identical *rrn* operon copies (16S-23S-5S rRNA), 49 tRNA genes, 4858 predicted protein-coding sequences, 2 Clustered Regularly Interspaced Short Palindromic Repeats (CRISPR) loci, and a set of CRISPR-associated (*cas*) genes. The high number of insertion sequence (IS) elements (over 200) suggests considerable genome plasticity.

The genome-based phylogeny of strain Shm1 was determined based on the comparative analysis of 120 ubiquitous single-copy proteins (Figure 3).

Their comparative analysis confirmed an earlier report that strain Shm1 is affiliated with the clade of *Methylococcus*-related, type Ib methanotrophs [11]. *Methyloterricola oryzae* 73a^T^ [21] and *Candidatus* Methyloumidiphilus alinensis [43] were identified as the two closest phylogenetic relatives of strain Shm1. The most closely related methanotroph with a complete genome sequence, however, was *Methylococcus capsulatus* Bath (Figure 2), which displayed 94.3% 16S rRNA gene sequence similarity to strain Shm1. Therefore, the genome of *Mc. capsulatus* Bath was used as a reference genome for comparative analysis.

### 3.4. Genome-to-GENOME comparison of Strain Shm1 and Mc. capsulatus Bath

Digital DNA–DNA hybridization and ANI values estimated for strain Shm1 and *Mc. capsulatus* Bath are 20.6% and 76.7%, respectively. The whole-genome alignment of these two methanotrophs is shown in Figure 4. Long regions of homology (64.9–72.3% identity), which are visualized by orange lines, include mostly the genes for rRNAs and tRNAs, C_1_ metabolism, general secretion pathways, hopanoid and fatty acid biosynthesis, some sensory proteins and a number of hypothetical proteins.

The comparison of general genome characteristics of strain Shm1 and *Mc. capsulatus* Bath is given in Table 1. The genome size (4.7 Mbp) and the number of rRNA operons (3) in strain Shm1 exceed those in *Mc. capsulatus* Bath (3.3 Mbp genome size and two copies of rRNA operons). The DNA G+C content in the moderately thermophilic *Mc. capsulatus* Bath (63.6 mol %) is significantly higher than in the mesophilic-to-psychrotolerant strain Shm1 (54.0 mol %).

Both methanotrophs possess an identical set of MMO-encoding genes and a highly similar array of C_1_-metabolism-related genes (Figure 5, Appendix A).

Two *pmoCAB* gene clusters encoding conventional particulate MMO and a single *mmoXYBZDC* gene cluster encoding soluble MMO were identified in both genomes. Methanol oxidation capabilities of these methanotrophs are explained by the presence of gene clusters encoding MxaFI- and XoxF-methanol dehydrogenases. Complete sets of genes for the function of the ribulose monophosphate pathway (RuMP), tetrahydromethanopterin—(H_4_MPT), and tetrahydrofolate (THF)—linked pathway were also present. Most of the genes for the serine cycle as well as those for the Calvin-Benson-Bassham (CBB) cycle were found in both genomes. However, a key difference in the genetic potential of the two methanotrophs was the presence of phosphoenolpyruvate carboxylase, which catalyzes the production of oxaloacetate from phosphoenolpyruvate, and absence of glycerate kinase in strain Shm1. Enzymes of CBB cycle fructose-1,6-bisphosphatase and sedoheptulose-1,7-bisphosphatase are not encoded in the genomes of *Mc. capsulatus* Bath and strain Shm1. Instead, the genomes of both methanotrophs encode transketolase (MCA3040, MCA3046, F6R98_00285) which was proposed to reversibly convert glyceraldehyde-3-phosphate to xylulose-5-phosphate, bypassing the typical ribose-5-phosphate to fructose-6-phosphate segment [15]. Therefore, both organisms have the genomic potential to operate all three known pathways for the assimilation of single-carbon compounds that are known to occur in various methylotrophs.

### 3.5. Genome-Encoded Adaptations to Oxygen Limitation

The genome of strain Shm1 encodes a complete electron transport chain and two ATP synthase operons, one of which was annotated as sodium-translocating NADH: quinone oxidoreductase. A large array of respiratory complexes that permit adaptation to a wide range of oxygen concentrations is also encoded in the genome of this methanotroph. All subunits of low-affinity aa_3_-type cytochrome *c* oxidase (F6R98_07605-F6R98_07620, F6R98_11440), which belongs to A-type family of the heme-copper oxidase (HCO) superfamily, were identified. This enzyme is also present in *Mc. capsulatus* Bath (MCA0879-MCA0883). Operon organization of cytochrome *c* oxidase in both methanotrophs is similar to that of the subtype b operon (*cox2*-*cox1*-*ctaB*-*ctaG*-*cox3*), which was identified to encode A1-type terminal oxidases [44]. However, in the genomes of strain Shm1 and *Mc. capsulatus* Bath, the *ctaB* gene (F6R98_11440) is located outside their cytochrome *c* oxidase operon. The Cox1 sequences of these two methanotrophs display high level of homology (93.15%). Phylogenetic analysis of Cox1 sequences suggested that *cox1* genes have been inherited vertically along the lineages of related methanotrophic bacteria (Figure 6).

The B- and C-family O_2_ reductases, which are considered to have high affinities for O_2_ [45], were not encoded in the genome of strain Shm1.

To explain the ability of strain Shm1 for growth at reduced oxygen tensions, we inspected its genome for the presence of *bd* terminal oxidases. These display, similar to C-type heme-copper oxidases, K_m_ values for O_2_ in the nanomolar range [44,46]. Indeed, we identified three copies (F6R98_00185-F6R98_00195; F6R98_03365-F6R98_03375; F6R98_15065-F6R98_15070) of *cydAB* operon encoding the cytochrome *bd*-I ubiquinol oxidase in the genome of strain Shm1, but only a single copy (MCA1105-MCA1106) in the genome of *Mc. capsulatus* Bath. Two of the three copies of the subunit I of cytochrome *bd*-I ubiquinol oxidase encoded in the genome of strain Shm1 (F6R98_00195 and F6R98_03365) display 87.48% amino acid identity and are homologous (77.44% and 75.91% identity) to the subunit I protein in *Mc. capsulatus* Bath. The third subunit I copy (F6R98_15070) encoded in the genome of strain Shm1 is highly divergent to the other two subunits (~40% amino acid identity) and its closest homologue is the subunit I protein encoded in the genome of *Methylobacter tundripaludum* (Figure 6). The presence of both low- and high-affinity terminal oxidases is common for most gammaproteobacterial methanotrophs (Table 2). Among the genomes analyzed in our study, only the genome of *M. buryatense* 5GB1C did not encode a *bd*-type oxidase.

Interestingly, the genome of *Mc. capsulatus* Bath encodes a terminal oxidase (MCA2396-MCA2397), which was annotated as *b(o/a)_3_*-type cytochrome *c* oxidase. The latter was shown to have high catalytic activity in oxidizing cytochrome *c*-551 with high affinity [47]. However, the protein encoded in the genome of *Mc. capsulatus* Bath displays similar level of identity to the corresponding sequences of *ba_3_* and *b(o)a_3_* oxygen reductases from *T. thermophilus* [34] and *Geobacillus stearothermophilus* [35], respectively, which complicates precise annotation of this protein. Terminal oxidases of this type are also encoded in the genomes of *Methylomonas* sp. LW13 and *M*. *buryatense* 5GB1C.

### 3.6. Nitrogen Metabolism

Genes encoding both the molybdenum–iron (Mo-Fe) and vanadium–iron (V-Fe) type nitrogenases were found in the genome of strain Shm1 (Figure 5). The genes encoding Mo-Fe-nitrogenase are assembled in 4 clusters: *nifBOQ*, *nifNE*, *nifMZWENX* and *nifXTKDHLA.* NifH in strain Shm1 is most closely related (94% amino acid identity) to that in *Methylomonas koyamae*. The V-Fe-nitrogenase, which is rarely found in methanotrophs, is encoded by the *vnfKGD* cluster. Phylogenetic analysis of the VnfD produced by strain Shm1 revealed high amino acid identity (94%) to the corresponding proteins from *Azotobacter* or *Azospirillum* species (Figure 7).

Strain Shm1 appears to be capable of reducing nitrate to ammonium by the activities of nitrate (NasAB) and nitrite (NirAB) reductases. The *norBC* genes encoding nitric oxide reductase were also found in the genome of strain Shm1.

### 3.7. Motility

Genes encoding flagella biosynthesis in the genome of strain Shm1 are located in three gene clusters (Figure 8, Appendix A).

The first cluster contains mainly the genes (a–e) responsible for the intracellular flagellum biosynthesis: (a) *fliPQR* encoding proteins associated with the type III secretion system; (b) *fliGMN* encoding proteins associated with the formation of the C-ring; (c) *fliF* encoding the proteins responsible for the formation of the P-ring; (d) *fliE* encoding the flagellar hook-basal body complex protein; and (e) *fliML* encoding the proteins involved in the formation of the hook and flagellin filament. The second cluster contains the flagellum elongation control genes *fliKTSD* and *flaG*. Finally, the third cluster is composed of the following genes (a–f): (a) Two copies of *motAB* encoding the motor complex; (b) *flgBCFG* encoding the basal body subunits; (c) *flgFNJ* and *flhAB* regulatory genes; (d) *flgD* encoding the flagellar basal-body rod modification protein; (e) *flgIH* encoding the proteins responsible for the formation of the L-ring; and (f) *flgLK* encoding the proteins responsible for connecting the hook with the flagellin filament and *flaB* gene encoding flagellin. One additional copy of the *flaB* gene is located outside of these gene clusters. This copy, however, is truncated and probably does not encode a functional protein. The *flaB* genes are important for maintaining the helical cell shape, and their damage leads to a change in shape and loss of motility [48,49]. Interestingly, the presence of *flgIH* genes potentially allows strain Shm1 to form both the periplasmic flagellum typical of spiral-shaped bacteria and the classical bacterial flagellum, with an “exoperiplasmic” flagellin filament. No motility-related genes were found in the genome of *Mc. capsulatus* Bath, although most of these genes were present in the genomes of other motile gammaproteobacterial methanotrophs (Figure 5). The latter, however, lack the *flgM* gene, an important regulator of flagella biosynthesis [50,51]. Phylogenetic analysis of FlaB sequence of strain Shm1 showed that it is highly divergent from flagellin sequences in other motile methanotrophs and displays highest identity (49%) to the corresponding protein sequences from members of the *Burkholderiaceae* (Appendix A).

### 3.8. Helical Cell Shape

In many cases, the cell shape in helical bacteria is determined by a modification of the peptidoglycan (PG) layer by PG-modifying enzymes [52]. For example, an array of such cell shape determinant (*csd1*–*scd6*) genes has been described for *Helicobacter pylori*. Curiously, these *csd* genes as well as the *ccmA* (curved cell morphology) gene encoding a bactofilin homolog are absent in the genome of strain Shm1. Most likely, the helical cell shape of our methanotroph strain is due solely to the presence of a periplasmic flagellum, which is the case in some spirochetes as, for example, *Borrelia burgdorferi* [49,53].

### 3.9. Insertion Sequences and Prophages

A total of 211 regions belonging to insertion sequences (ISs) were identified in the genome of strain Shm1. By contrast, the genome of *Mc. capsulatus* Bath contains only 61 ISs. The diversity of IS types in strain Shm1 is almost two times greater than that in *Mc. capsulatus* Bath. Most ISs in both methanotrophs belong to the IS3 and IS5 families. The third most prevalent IS group in the genome of strain Shm1 is the family IS21. ISs of the families IS110, IS21, IS30, IS4, IS630, IS66, and IS91 were found in the genome of strain Shm1, but were absent in *Mc. capsulatus* Bath. Both IS abundance and diversity in the genome of strain Shm1 suggests that intra-genomic rearrangements and horizontal gene transfer events may have been important for the evolution of this spiral-shaped methanotroph.

Five regions associated with prophages were found in the genome of strain Shm1. The set of genes in these regions is incomplete; accordingly, a virulent phage cannot be formed on their basis. However, these sites work as ISs and may play an important role in the intra-genomic gene rearrangements. Interestingly, two regions associated with prophages were revealed in the *Mc. capsulatus* Bath genome, with a set of genes sufficient to form a complete phage capsid.

### 3.10. Signaling Systems

Analysis of signaling machineries in strain Shm1 was performed in order to reveal the major classes of sensor proteins, which play a significant role in its response to environmental stimuli. This methanotroph possesses a wide range of sensor proteins, which belong to six major classes, i.e., histidine kinases (43 proteins), methyl-accepting chemotaxis proteins (29), proteins with diguanylate cyclase (39) and/or c-di-GMP-specific phosphodiesterase activity (15), membrane components of the sugar phosphotransferase system (2), and predicted serine/threonine protein kinases (5) (Figure 9).

Functional annotation of the histidine kinases (HKs) in strain Shm1 was complicated due to low sequence identities (<30% on amino acid level) with signaling proteins of model bacteria such as *E. coli* and *B. subtilis*. These are commonly used as reference organisms for describing metabolic and signaling pathways in less-studied organisms. Thus, the analysis of HKs identified in the genome of strain Shm1 was based on the alignment of specific sequence motifs (H-, N-, G1-, G2-, and -F boxes) that are known to be conserved across the kinase core domains of HKs [54,55]. Our phylogenetic analysis separated the HK gene families into three major branches (Appendix A). The majority of HK proteins (35) in strain Shm1 were represented by Type I HKs that are involved in the response to a wide range of stimuli. One HK belonged to Type III group, which includes sensors responsive to nitrate, nitrite (NarX, NarQ), and glucose-6-phosphate (UhpB) [56]. Strain Shm1 lacked Type II HKs which are responsive to sensing di- and tricarboxylates (CitA, DcuS). It, however, possessed seven CheA HKs that directly interact with methyl-accepting chemotaxis proteins (MCPs) as well as seven specialized response regulators CheYs, which directly modulate motion of the flagellar motor [57]. In addition to CheA, six gene clusters in the genome of strain Shm1 contain another core chemotaxis proteins such as CheW, CheY, and MCP, which are present in all chemotaxis systems [58]. The methyltransferase CheR and methylesterase CheB were identified in five gene clusters, while CheZ was present only in one set of chemotaxis proteins. A similar pattern of HKs was identified in the genome of *Mc. capsulatus* Bath. However, among 16 HKs only one protein was annotated as CheA.

The genome of strain Shm1 encodes 29 methyl-accepting chemotaxis proteins (MCPs), most of which are related to class I MCPs. Based on the types of ligand-binding domains (LBD) and membrane topology, 15 MCPs belong to cluster I with four-helix bundle (4HB) and Per-Arnt-Sim (PAS) domains. One MCP contains a NIT domain, which recognizes specifically nitrate and nitrite [59,60]. One MCP of Class IV was identified to have a Protoglobin domain, a member of the globin superfamily, which acts as an oxygen sensor and relates to aerotaxis proteins [61,62]. Two proteins annotated as MCPs included only a MCPsignal domain, eight lacked LBD, and two proteins were arranged as dimers lacking HAMP domain and LBD. Notably, the genome of *Mc. capsulatus* Bath encodes only two MCPs with HAMP-MCPsignal domain structure. MCPs with Protoglobin domain are not present in this methanotroph.

The genome of strains Shm1 encodes 39 diguanylate cyclases (proteins with the conserved GGDEF domain) that respond to environmental or intracellular signals by synthesizing the second messenger c-di-GMP [63,64]. A total of 30 c-di-GMP-specific phosphodiesterases (EAL and HD-GYP domains) that affect a variety of c-di-GMP-regulated systems were also identified. The genome of strains Shm1 encodes 18 tandem GGDEF-EAL domain-containing proteins that may possess diguanylate cyclase and/or phosphodiesterase enzymatic activities. Proteins with GGDEF (5), GGDEF-EAL (11), HD-GYP (1) domains are also present in *Mc. capsulatus* Bath.

We also examined the repertoires of signal transduction systems encoded in the genomes of several representative motile methanotrophs, such as *Methylomonas methanica* MC09, *Methylomonas* sp. LW13, *Methylobacter tundripaludum* SV96, and *Methylomicrobium buryatense* 5GB1C (Table 3). The most abundant types of sensory proteins encoded in the genomes of these bacteria include response regulators, histidine kinases, methyl-accepting chemotaxis proteins, and diguanylate cyclases. In general, the number of signal transduction systems encoded in the genomes of motile gammaproteobacterial methanotrophs was comparable to that in strain Shm1.

## 4. Discussion

Three years after the original description of *Candidatus* Methylospira mobilis [11], this unusual spiral-shaped methanotroph was obtained in pure culture. Apparently, the association with satellite bacteria is not obligately required for growth of *Methylospira mobilis*. Major difficulties in obtaining an axenic culture of this spiral-shaped methanotroph originated from its inability to develop on solid media and its preference for growth under micro-oxic conditions. The original description characterizes *Candidatus* Methylospira mobilis as an obligately micro-aerobic bacterium that is incapable of growth under fully oxic conditions. By contrast, the isolate obtained in our study, strain Shm1, was able to grow under aerobic conditions. Since 16S rRNA gene sequence of strain Shm1 did not fully match the sequence obtained by Danilova et al. [11], the employed isolation approach could have resulted in selecting a phenotype with somewhat higher tolerance to oxygen. As shown in our incubation studies, strain Shm1 was capable of development in a wide range of O_2_ concentrations, including fully aerobic conditions, although highest growth rates were detected at reduced (5%) oxygen concentrations. Similar preferences for reduced O_2_ concentrations were earlier reported for several gammaproteobacterial methanotrophs isolated from lake sediments, such as *Methylosoma difficile* and *Methyloglobulus morosus* [65,66]. *Candidatus* Methyloumidiphilus alinensis, one of the closest phylogenetic relatives of strain Shm1, was also detected at the oxycline of a humic lake in Finland, at a depth of 3 to 4.5 m [43].

Growth in micro-oxic habitats requires possession of *bd*-type oxidases known for their high affinity to oxygen [46]. A set of terminal oxidases encoded in the genome of strain Shm1 included three copies of genes encoding *bd*-oxidases and only a single copy of the gene for low-affinity cytochrome *c* oxidase. Obviously, this reflects the preference of *Methylospira mobilis* for growth under micro-oxic conditions. In addition, a large body of evidence suggests that cytochrome *bd* contributes to bacterial resistance against oxidative and nitrosative stress [67,68]. Therefore, *bd*-oxidases of strain Shm1 may have a protective function during its growth under fully aerobic conditions.

Another important trait of *Methylospira mobilis*, which makes this methanotroph well-suited for life in wetlands, is its capability of a rapid movement in water-saturated, heterogeneous habitats, with the possibility of positioning its cells in an optimal range of O_2_ and substrate concentrations. This capability is defined by the complete set of motility genes present in the genome of *Methylospira mobilis* and an impressive array of signal transduction proteins. Our analysis showed that more or less complete sets of motility-related genes are also present in the genomes of other motile gammaproteobacterial methanotrophs (Figure 5). The proteins encoded in genomes of these bacteria, however, display only a very low homology to those in strain Shm1, thereby suggesting major differences in the motility machineries between these methanotrophs.

Genes encoding nearly two hundred signal transduction proteins were identified in *Methylospira mobilis*. The total number of regulatory proteins encoded in the genome of a given organism positively correlates with its genome size [69]. Indeed, the number of signal transduction proteins in strain Shm1 significantly exceeds that in *Mc. capsulatus* Bath but is comparable to those in other motile gammaproteobacterial methanotrophs (Figure 9, Table 3). Histidine kinases are the most numerous and most diverse membrane receptors encoded in bacterial genomes. Accordingly, they control the greatest variety of cellular responses [56]. Based on both sequence differences and organization of the H-box and kinase domains, the majority of HKs in the genomes of strain Shm1 and *Mc. capsulatus* Bath belongs to type I HKs [55]. Type I HKs perceive a big variety of signal molecules including phosphate; oxygen and/or hydrogen peroxide; heavy metals, such as Cu2^+^/Ag^+^ or Zn^2+^ and Pb^2+^; glutamine; and trimethylamine N-oxide. These sensory proteins are responsible for the cellular response to envelope and osmotic stress, including the adjustment of the magnitude of K^+^ gradient and quorum sensing. Besides numerous type I HKs, strain Shm1 possessed seven CheA proteins, which are directly involved in the signal processing in chemotaxis pathway [57,70]. Another big group of signaling proteins in both methanotrophs includes diguanylate cyclases (proteins with the conserved GGDEF domains). These respond to intracellular signals by synthesizing the second messenger c-di-GMP. Cyclic di-GMP has been shown to regulate a variety of systems, including motility, protein and polysaccharide secretion, cell division and biofilm formation, and intracellular signals [63,64].

The composition of signal transduction systems is quite similar in *Methylospira* and *Methylococcus*. However, the genome of *Mc. capsulatus* Bath encodes very few MCPs while the number of signaling proteins present in strain Shm1 is one order of magnitude higher. MCPs recognize various signals and transmit this information to CheA, which ultimately controls the direction of flagellar motor rotation [71]. Therefore, genes encoding MCPs were mainly found in genomes of motile microbes [72]. Overall, the repertoire of signal transduction proteins in the studied organisms reflects their lifestyles. *M. capsulatus* Bath possesses signaling machineries for passive sensing of environmental variables, while chemotaxis systems in strain Shm1 allow active selection of loci with most favorable conditions.

One additional, environmentally important characteristic of *Methylospira*-like bacteria is their ability to fix dinitrogen. Mo-Fe-nitrogenase is widespread among bacteria and is present in many methanotrophs. By contrast, V-Fe-nitrogenase is quite rare, which is likely due to the bioavailability of trace elements [73]. Earlier, the genes encoding V-Fe-nitrogenase were found in the genomes of alphaproteobacterial methanotrophs of the genus *Methylocystis* [74,75], but this enzyme was not previously detected among gammaproteobacterial methanotrophs. Phylogenetic analysis of the *vnfD* gene showed that, while representatives of the genus *Methylocystis* inherited the V-Fe-nitrogenase genes due to vertical gene transfer, the *vnfD* gene cluster in the genome of strain Shm1 is probably a result of a lateral gene transfer from *Azospirillum* or *Azotobacter* species (Figure 6). Curiously, all currently described methanotrophs with V-Fe-nitrogenase were isolated from oligotrophic wetlands. Perhaps the ability to synthesize two different nitrogenase types represents an important adaptation to low trace element availability in the environment.

In summary, our analysis of strain Shm1 identified a number of genome-encoded traits, which define the lifestyle of a spiral-shaped methanotroph and a number of its adaptations to life in wetlands. The complete genome sequence, along with the axenic culture of this methanotroph, opens the possibility for further taxonomic description of a novel genus and species and the valid publication of its name, *Methylospira mobilis*.

## Figures and Tables

**Figure 1 microorganisms-07-00683-f001:**
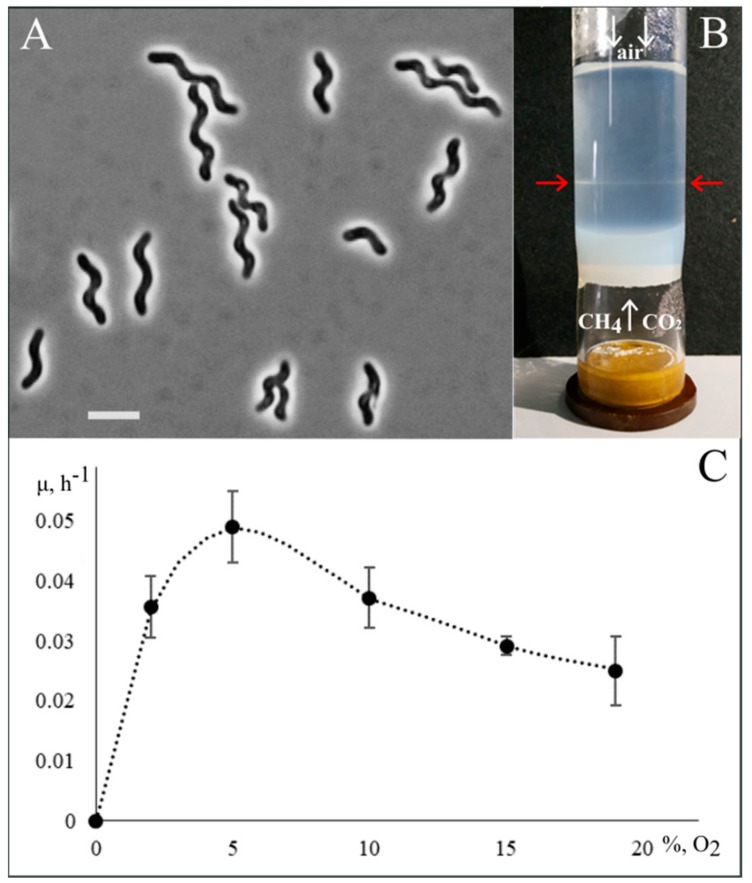
Cell morphology and physiology of strain Shm1. (**A**) Phase-contrast micrograph of cells of strain Shm1; bar, 2 µm. (**B**) Growth of strain Shm1 in the opposing gradients of methane and oxygen. White arrows indicate fluxes of gases into semi-solid medium. Red arrows point to the growth band formed by this methanotroph. (**C**) Specific growth rates of strain Shm1 at different oxygen concentrations.

**Figure 2 microorganisms-07-00683-f002:**
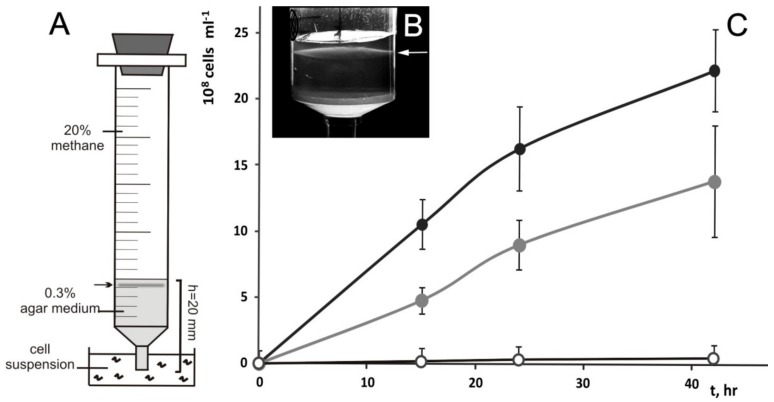
Chemo-tactic responses of strain Shm1. (**A**,**B**) The experimental setup used to determine chemo-tactic response of strain Shm1. Arrows point to the subsurface faint bands of cells, which are formed after 24 h of incubation with methane or methanol in the headspace of syringes. (**C**) Dynamics of cells accumulation in a subsurface agar layer during incubation with methane (black circles) or methanol vapors (gray circles) in the headspace of syringes versus control incubations (white circles).

**Figure 3 microorganisms-07-00683-f003:**
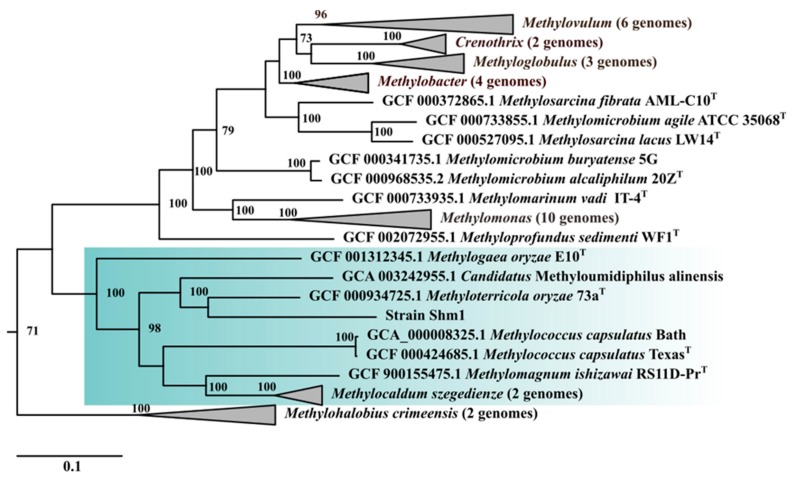
Genome-based phylogeny of the spiral-shaped methanotroph. Phylogenomic tree showing the position of strain Shm1 among other gammaproteobacterial methanotrophs based on the comparative sequence analysis of 120 ubiquitous single-copy proteins. The clade of *Methylococcus*-related type Ib methanotrophs is highlighted by green. The tree was constructed using the Genome Taxonomy Database toolkit [37], release 04-RS89. The significance levels of interior branch points obtained in maximum-likelihood analysis were determined by bootstrap analysis (100 data re-samplings). Bootstrap values of > 70% are shown. The root is composed of 16 genomes of type IIa methanotrophs of the family *Methylocystaceae*. Bar, 0.1 substitutions per amino acid position.

**Figure 4 microorganisms-07-00683-f004:**
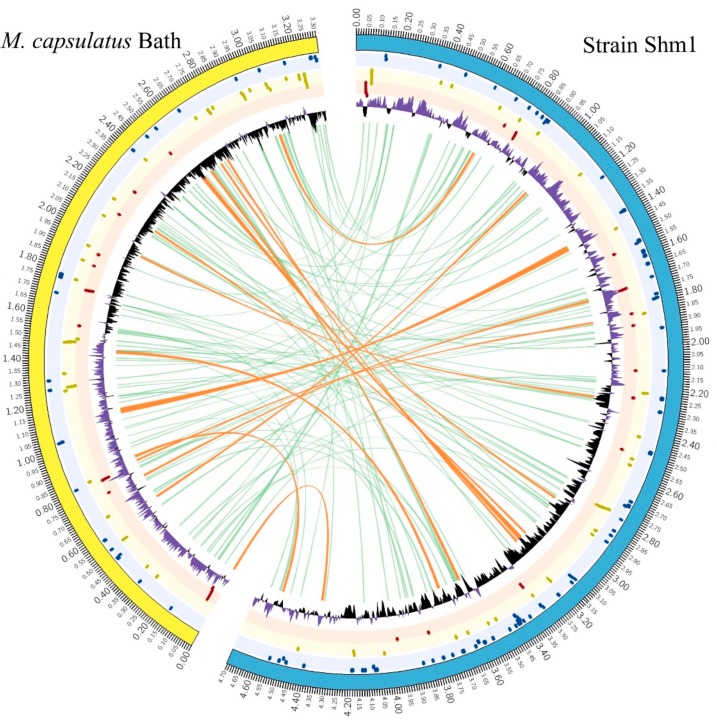
Comparison of the genomes of strain Shm1 and *Mc. capsulatus* Bath. The whole-genome alignment is visualized by green (short regions) and orange (long regions) curves. Concentric circles from the outermost to the center include: (1) GC skew (purple > 0, black < 0), (2) sensory genes (blue dots), (3) C_1_ metabolism genes (yellow dots), (4) oxygen metabolism genes (red dots).

**Figure 5 microorganisms-07-00683-f005:**
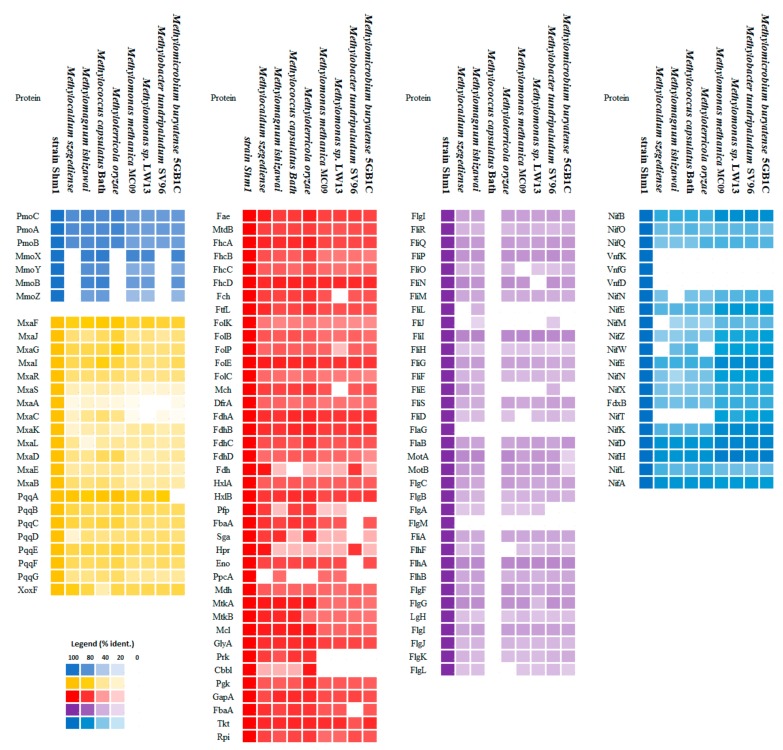
Heatmap showing the presence of selected groups of genes in the genomes of strain Shm1-related methanotrophs. Methane monooxygenase (MMO)-encoding genes (green), methanol dehydrogenase-encoding genes (brown), selected genes of C_1_ metabolism (red), nitrogenase-encoding genes (blue) and motility genes (purple) are shown. Color intensity reflects homology of the respective proteins encoded in the genome of strain Shm1 to those in other methanotrophs.

**Figure 6 microorganisms-07-00683-f006:**
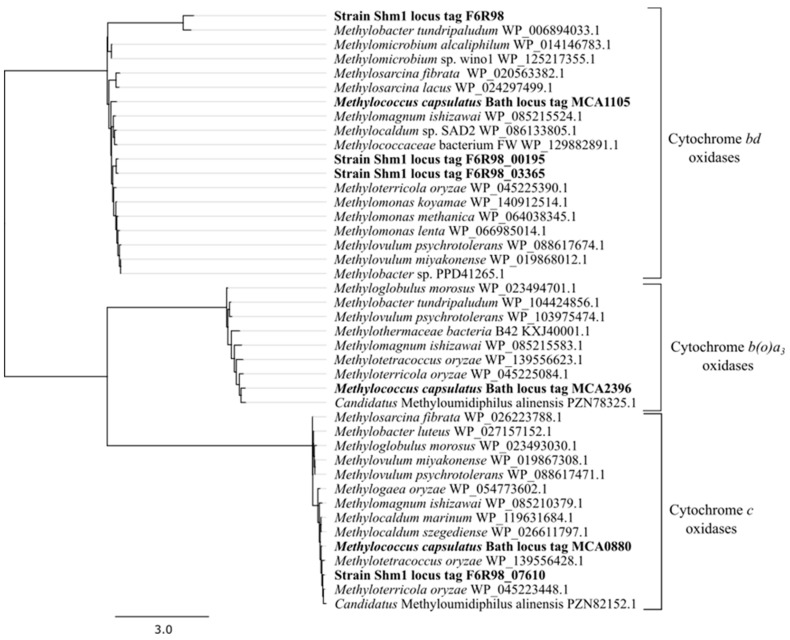
Phylogenetic analysis of respiratory proteins encoded in the genome of strain Shm1. Cytochrome *c* and *bd* oxidases are shown. Proteins from strain Shm1 and *Mc. capsulatus* Bath are indicated in bold characters.

**Figure 7 microorganisms-07-00683-f007:**
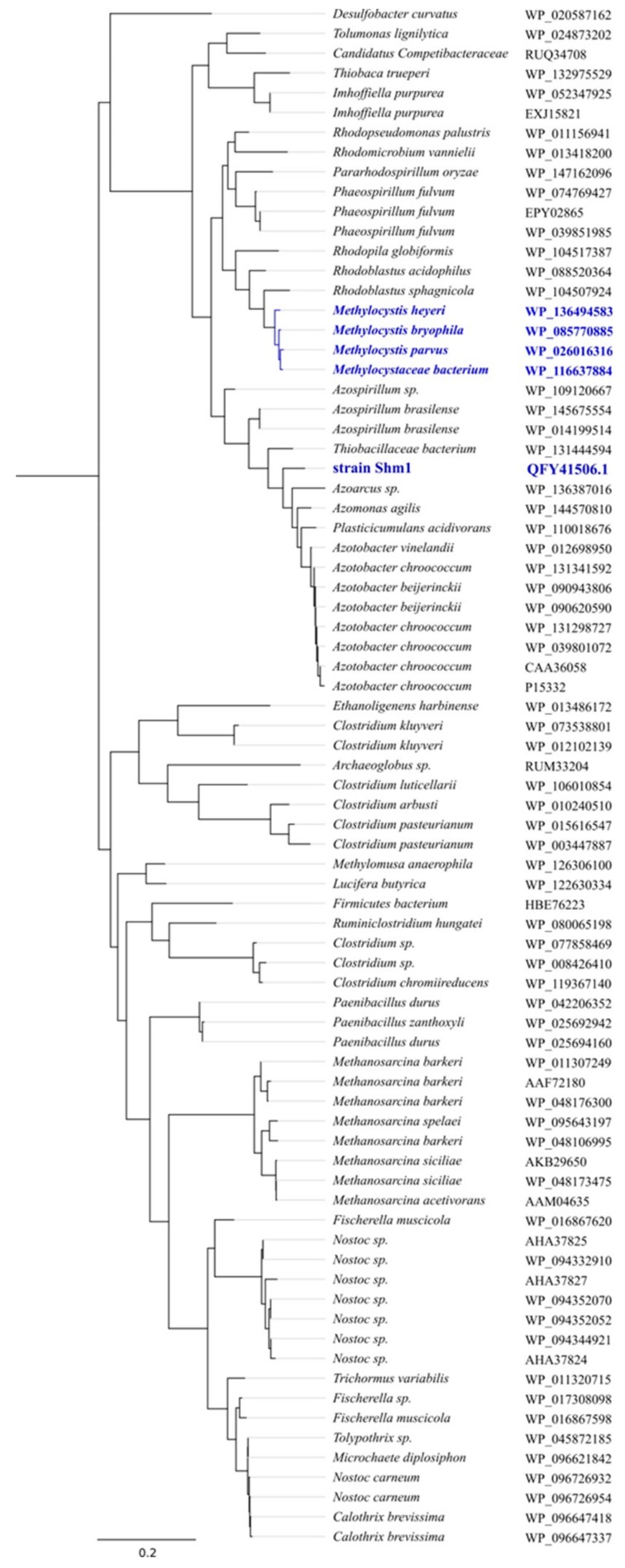
Phylogenetic analysis of the VnfD from strain Shm1. The tree was constructed based on 459 amino acid positions of VnfD sequences using the Fast Tree package and approximately maximum-likelihood method. The root is composed of 41 AnfD sequences. Proteins from methanotrophic bacteria are shown in blue. Bar, 0.2 substitutions per amino acid position.

**Figure 8 microorganisms-07-00683-f008:**
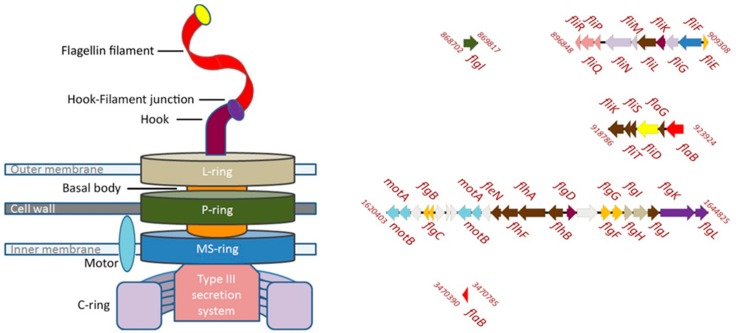
Scheme showing the motility apparatus (left panel) and the organization of the genes encoding this apparatus in the genome of strain Shm1 (right panel). Color designations of the genes and the corresponding components of the motility apparatus are the same on both panels. The genes encoding flagellin (marked red), hook-filament junction (purple), hook (tyrian purple), L-ring (sisal), P-ring (green), MS-ring (blue), C-ring (melrose), type III secretion system (petite orchid), and basal body (orange) are shown. Regulation genes are marked brown.

**Figure 9 microorganisms-07-00683-f009:**
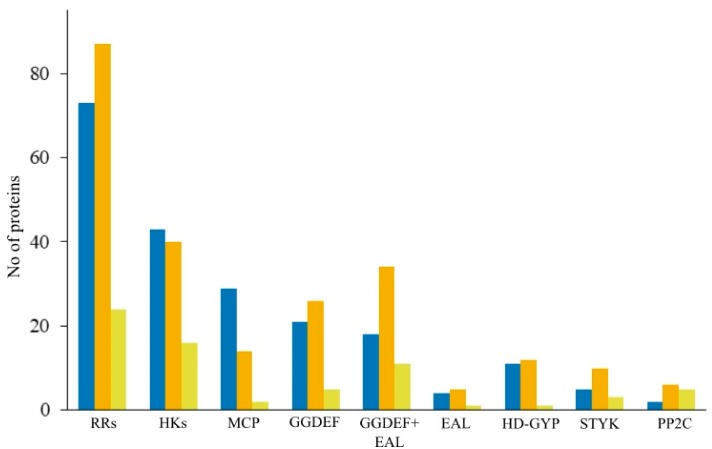
The number of signal transduction proteins encoded in the genomes of strain Shm1 (blue), *M. methanica* MC09 (orange) and *M. capsulatus* Bath (yellow). Abbreviations: RRs—response regulators, HKs—histidine kinases, MCP—methyl-accepting chemotaxis proteins, GGDEF—diguanylate cyclases (proteins with GGDEF domain), EAL, HD-GYP—c-di-GMP-specific phosphodiesterases (proteins with EAL or HD-GYP domains), STYK—Ser/Thr/Tyr protein kinases, PP2C—Ser/Thr/Tyr phosphoprotein phosphatases.

**Table 1 microorganisms-07-00683-t001:** General genome features of strain Shm1 and *Mc. capsulatus* Bath.

Genome Characteristics	Strain Shm1	*Mc. capsulatus* Bath
Accession number	CP044205	AE017282
Size (MB)	4.7	3.3
Contigs	1	1
G+C content (mol %)	54.0	63.6
Coding sequences	4858	3167
rRNAs (5S, 16S, 23S)	3, 3, 3	2, 2, 2
tRNAs	49	51
CRISPR loci	2	2
pMMO operon	2	2
sMMO operon	1	1
IS elements	211	61
Prophages	5 (incomplete)	2 (complete)

**Table 2 microorganisms-07-00683-t002:** Terminal oxidases encoded in the genomes of strain Shm1, *Mc. capsulatus* Bath and other representative gammaproteobacterial methanotrophs.

Microorganism (Genome Accession Number)	Total Number of Terminal Oxidases			
	Low-Affinity		High-Affinity	High-/Low-Affinity
	aa_3_	bo_3_	bd	b(o)a_3_/ba_3_
Strain Shm1 (CP044205)	1 *	none	3	none
*Methylococcus capsulatus* Bath (AE017282.2)	1	none	1	1
*Methylomonas methanica* MC09 (NC_015572.1)	2	1	2	none
*Methylomonas* sp. LW13 (NZ_CP033381.1)	1	none	2	1
*Methylobacter tundripaludum* SV96 (AEGW00000000.2)	2	none	2	none
*Methylomicrobium buryatense* 5GB1C (NZ_CP035467.1)	1	1	none	1

* The information regarding location of the corresponding genes is given in Appendix A.

**Table 3 microorganisms-07-00683-t003:** Signal transduction systems encoded in the genomes of strain Shm1, *Mc. capsulatus* Bath, and several representative motile methanotrophs.

Microorganism	Total Proteins	RRs	HisK	MCP	GGDEF	GGDEF +EAL	EAL
Strain Shm1	4043	73	43	29	21	18	4
*Mc. capsulatus* Bath	2959	24	16	2	5	11	1
*M. methanica* MC09	4403	87	40	14	26	34	5
*Methylomonas* sp. LW13	4609	101	51	20	23	30	4
*M. tundripaludum* SV96	4080	75	46	22	20	26	4
*M. buryatense* 5GB1C	4254	102	50	18	21	37	4

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
