# Peer review of "Thriving in Wetlands: Ecophysiology of the Spiral-Shaped Methanotroph *Methylospira mobilis* as Revealed by the Complete Genome Sequence"

_microorganisms, 2019, doi:10.3390/microorganisms7120683_

Round 1

Reviewer 1 Report

This paper describes an exciting news that a pure culture of a novel methanotroph has been obtained and its genome has been sequenced. However, the approach to genome analysis is very disappointing. What is the ground for comparing this novel genome to a single other methanotroph genome? Methylococcus capsulatus is arbitrarily selected. Indeed this was the first methanotroph genome to be analyzed, but that happened 15 years ago, and the team involved were not experts in methanotrophy. The novel genome needs to be compared to the genomes of key methanotroph taxa, including motile methanotrophs such as Methylomonas. I understand that some genomes in the databases are of low quality, but there many genomes of very high quality including finished genomes. Without such comprehensive analysis the manuscript does not go much beyond genome announcement. A very limited attempt is presented in Fig 4, and it needs to be expanded to include other Methylococcaceae. For examp[le, how the flagellum functions in Methylomonas diffrer from the ones presented in Fig. 7? On another hand, low-quality genomes such as M. orysae are not very useful as they appear to miss many essential genes. I suggest that the authors revise the manuscript to include these important comparisons, especially for the functions they feel are important for the organism to cope with microoxic conditions in wetlands, its cell shape, motility, signal transduction etc.

Author Response

We thank the referee for these comments.  Our responses are given below:

Comment: What is the ground for comparing this novel genome to a single other methanotroph genome? Methylococcus capsulatus is arbitrarily selected. Indeed this was the first methanotroph genome to be analyzed, but that happened 15 years ago, and the team involved were not experts in methanotrophy. The novel genome needs to be compared to the genomes of key methanotroph taxa, including motile methanotrophs such as Methylomonas. Fig 4 needs to be expanded to include other Methylococcaceae. On another hand, low-quality genomes such as M. orysae are not very useful as they appear to miss many essential genes. I suggest that the authors revise the manuscript to include these important comparisons, especially for the functions they feel are important for the organism to cope with microoxic conditions in wetlands, its cell shape, motility, signal transduction etc.

Response: The logic of comparing the genomes of Methylospira and Methylococcus is sound and easy to follow: these are two closely related but phenotypically different methanotrophs for which finished genome sequences are available. This is a backbone of our manuscript and we would like to keep it. However, we fully agree that the comparison of genes related to motility, signal transduction or adaptations to micro-oxic conditions should have been done by using several representative genomes of motile methanotrophs. We now do so by including four additional, good-quality genomes of Methylomonas methanica MC09, Methylomonas sp. LW13, Methylobacter tundripaludum SV96 and Methylomicrobium buryatense 5GB1C in the comparison. The genome of M. orysae has been omitted. Fig 4 has been replaced with its revised version. We have also included three additional tables (Tables 2 and 3, Supporting Table S2) in order to present the information regarding terminal oxidases and signal transduction systems encoded in the genomes of Methylospira, Methylococcus and four other representative gammaproteobacterial methanotrophs. Reviewer is absolutely right that the number of signal transduction systems in Methylospira is comparable to that in other motile methanotrophs. This is now reflected in the newly prepared Table 3 and revised Fig. 9. The corresponding statement in the Abstract has also been corrected.

Comment: For examp[le, how the flagellum functions in Methylomonas diffrer from the ones presented in Fig. 7?

Response: We have invested several days in comparing the arrangement of gene clusters encoding motility in Methylospira and other motile gammaproteobacterial methanotrophs but were unable to come up with a clear picture. Apparently, the arrangement of these gene clusters may vary even within one particular genus. A separate study is needed in order to address this question. However, we’ve noticed that the XXX is present only in the genome of Methylospira. In addition, the flagellin in Methylospira is highly divergent from that in all other methanotrophs. This info is now demonstrated in Supplementary Fig. S2.

Reviewer 2 Report

In this manuscript, Oshkin and co-workers sequenced and analyzed the genome of Methylospira mobilis, the first spiral-shaped metanotroph reported (Danilova et al., 2016, ISME J. 10: 2734-43). This bacterium was isolated from a peat sample and it was originally named as Candidatus Methylospira mobilis due to the impossibility of obtaining pure cultures of this bacterium. In this study, the authors managed to obtain pure cultures of this metanotroph and re-named this bacterium as Methylospira mobilis strain Shm1. Genome comparative analysis revealed that the genome of Shm1 is considerably larger than that of the model methanotroph M. capsulatus Bath. Remarkably, the genome of Shm1 possesses a considerably high number of chemoreceptor encoding genes.

I have several comments that will enhance the impact of this manuscript.

Previous research by the authors revealed that Methylospira mobilis exhibited CH4-tactic responses. However, these assays were not conducted with the purified strain but rather in the presence of Magnetospirillum-like bacteria that were tightly attached to this methanotroph. Furthermore, phenotypic characterization of Shm1 (e.g. aerobic growth) and the analysis of its 16S rRNA revealed some differences with the original Candidatus Methylospira mobilis. For this reason, and given that the authors have optimized a protocol to measure the chemotaxis properties of this bacterium (Danilova et al., 2016, ISME J. 10: 2734-43), the authors should include new data on the chemotactic properties of strain Shm1. Also, given the number of chemoreceptor proteins encoded in the genome of Shm1, these experiments should not be restricted to CH4- and O2-chemotaxis assays but should also include the chemotactic properties to (or away from) additional compounds (e.g. compounds that can be present in the original environmental niche were this bacterium was originally isolated). Lines 374-376: it is highly surprising that after the analysis of the genome of Shm1 the authors identified 7 CheA autokinases encoding genes but only one CheY response regulator encoding gene. The authors should re-analyze this data to confirm that identified genes do encode CheA proteins. The authors also should identify the remaining chemotaxis-related genes (e.g. cheW, cheR, cheB, etc) in order to determine how many chemosignaling pathways are present in the genome of Shm1 - not all chemosignaling pathways modulate chemotaxis.

Specific comments:

Sub-sections 2.1. (“Isolation procedure”) and 2.2. (“Cultivation in a range of oxygen concentrations”) could be considerably reduced in length.

Author Response

We thank the referee for these comments. Our responses are given below.

Comment: Previous research by the authors revealed that Methylospira mobilis exhibited CH4-tactic responses. However, these assays were not conducted with the purified strain but rather in the presence of Magnetospirillum-like bacteria that were tightly attached to this methanotroph. Furthermore, phenotypic characterization of Shm1 (e.g. aerobic growth) and the analysis of its 16S rRNA revealed some differences with the original Candidatus Methylospira mobilis. For this reason, and given that the authors have optimized a protocol to measure the chemotaxis properties of this bacterium, the authors should include new data on the chemotactic properties of strain Shm1. Also, given the number of chemoreceptor proteins encoded in the genome of Shm1, these experiments should not be restricted to CH4- and O2-chemotaxis assays but should also include the chemotactic properties to (or away from) additional compounds (e.g. compounds that can be present in the original environmental niche were this bacterium was originally isolated).

Response: Following this recommendation, we have performed a series of chemotactic experiments with pure culture of Methylospira mobilis. The results are shown in Figure 2. Very fast movement was observed towards methane and methanol, while no movement was detected in control incubations. We find it difficult, however, to follow the recommendation of including “additional compounds” in this analysis. Methylospira mobilis is an obligate methanotroph, which utilizes methane and methanol only. Again, detailed investigation of chemotactic machinery in this bacterium may take up to several years and is clearly beyond the scope of this study.

Comment: Lines 374-376: it is highly surprising that after the analysis of the genome of Shm1 the authors identified 7 CheA autokinases encoding genes but only one CheY response regulator encoding gene. The authors should re-analyze this data to confirm that identified genes do encode CheA proteins. The authors also should identify the remaining chemotaxis-related genes (e.g. cheW, cheR, cheB, etc) in order to determine how many chemosignaling pathways are present in the genome of Shm1 - not all chemosignaling pathways modulate chemotaxis.

Response: This is our mistake; we apologize. We have carefully re-analyzed the array of chemotaxis-related genes in Methylospira mobilis. Of course, the genome encodes 7 CheA autokinases and 7 CheY response regulators. The info about remaining chemotaxis-related genes is also presented now.

Comment: Sub-sections 2.1. (“Isolation procedure”) and 2.2. (“Cultivation in a range of oxygen concentrations”) could be considerably reduced in length.

Response: Done.

Round 2

Reviewer 1 Report

I am satisfied with the revisions.

Reviewer 2 Report

The authors have satisfactorily addressed all the comments raised by this reviewer. I have no additional comments on this manuscript.